# Bridge crack detection based on improved single shot multi-box detector

**Guanlin Lu, Xiaohui He⬥*, Qiang Wang, Faming Shao, Jinkang Wang, Qunyan Jiang**

Department of Mechanical Engineering, College of Field Engineering and Army Engineering University, PLA, Nanjing, China

* gcbhxh314@163.com

## Abstract

Owing to the development of computerized vision technology, object detection based on convolutional neural networks is being widely used in the field of bridge crack detection. However, these networks have limited utility in bridge crack detection because of low precision and poor real-time performance. In this study, an improved single-shot multi-box detector (SSD) called ISSD is proposed, which seamlessly combines the depth separable deformation convolution module (DSDCM), inception module (IM), and feature recalibration module (FRM) in a tightly coupled manner to tackle the challenges of bridge crack detection. Specifically, DSDCM was utilized for extracting the characteristic information of irregularly shaped bridge cracks. IM was designed to expand the width of the network, reduce network calculations, and improve network computing speed. The FRM was employed to determine the importance of each feature channel through learning, enhance the useful features according to their importance, and suppress the features that are insignificant for bridge crack detection. The experimental results demonstrated that ISSD is effective in bridge crack detection tasks and offers competitive performance compared to state-of-the-art networks.

**Data Availability Statement:** Article data comes from public data sets(SDNET dataset and CCIC dataset). SDNET: https://digitalcommons.usu.edu/all_datasets/48/ CCIC: https://data.mendeley.com/datasets/5y9wdsg2zt/2.

## 1. Introduction

As a fundamental component of the transportation system, the bridge not only takes responsibility for transporting items but also ensures the safety of the transport personnel. However, bridges are prone to various types of damage owing to natural or human factors. Among them, deck cracks are a common problem in bridge services. Cracks in a bridge accelerate the speed of corrosion of the armature, resulting in deterioration of the bridge structure [1]. Furthermore, the presence of cracks affects the integrity, durability, and seismic performance of a bridge and considerably reduces its quality [2, 3]. To maintain the healthy state of bridges, it is important for the engineering community, national government administrative services, and bridge construction companies to detect and repair cracks in a timely manner.

The development of bridge crack-detection methods has been relatively slow. Traditional manual detection is not only time-consuming and laborious but also has many unsafe factors. The bridge inspection vehicle is a special vehicle that can provide a working platform for

**Funding:** This research was funded by the National Natural Science Foundation of China (grant number: 61671470) and the Key Research and Development Program of China (grant number: 2016YFC0802900).

**Competing interests:** No conflict of interest.

bridge inspection personnel during the inspection process and is equipped with bridge inspection instruments for flow inspection and/or maintenance operations. However, its utility is limited by its high production cost and complex manufacturing process.

Non-destructive testing technology has been widely used in the field of bridge crack detection. Common non-destructive testing methods include optical fiber sensing [4], ultrasonic detection [5], and acoustic emission detection [6]. However, these non-destructive methods have some limitations. Optical fiber sensing technology requires the laying of optical fibers, which is expensive. Acoustic detection technology is only suitable for detecting cracks in a single direction of a bridge deck with a small detection range. Acoustic emission detection technology can only detect cracks that are being generated at present and cannot detect cracks that have previously formed. Therefore, the high detection cost, limited working conditions, and inefficient detection speed limit the traditional detection methods based on manual detection or instrument information characteristic analysis, and it is important to devise a new technical means to carry out real-time and efficient bridge crack detection.

Computer vision technology has improved rapidly with the rapid development of computer automation. As an important research topic in the field of computer vision, the main task of object detection is to locate a target of interest in an image and accurately judge the specific category and location of the target. In recent years, object detection has been widely used in intelligent video surveillance, fault detection, medical treatment, and other fields. Many scholars have proposed different types of object detection models for crack detection. Li et al. [7] presented a model based on a support vector machine (SVM) to detect bridge cracks. Nishikawa et al. [8] used several simple image filters to design a multi-sequential image filter. Wang et al. [9] presented a model based on mathematical morphology for detecting cracks in steel. Cha et al. [10] combined the Hough transform with an SVM to detect cracks. These methods mainly use manually extracted features to detect cracks. Compared with the traditional crack detection technology, it improves the detection accuracy and speed. However, the results of these methods are affected by human subjective factors in feature processing, such as people's professional ability, grasp of standards, and other complex factors.

In recent years, convolutional neural networks (CNN) have made significant progress in object detection [11–14]. Many researchers have applied CNN to crack detection. Chen et al. [15] presented a network called NB-CNN, which combines a CNN with naïve Bayes data fusion to detect cracks. This algorithm can only recognize the location of cracks with a low detection accuracy. To achieve higher detection accuracy, Cha et al. [16] proposed a network based on faster R-CNN [17]; however, a large number of parameters affect the network detection speed. Dung et al. [18] proposed a crack model based on a fully convolutional network (FCN) for crack detection because the complex environmental noise that influences detection accuracy is not high. Various algorithms have been applied to crack detection, but the data samples used for model training are often collected in an ideal environment and manually modified in the labeling process. During the actual detection of the model, the interference of the environment (such as illumination, occlusion, jitter, etc.) will cause the domain offset of the input image, reduce the model to extract useful crack features, and increase the difficulty of the model feature processing. The output results are mixed with too many irrelevant features, limiting the model's detection accuracy. Therefore, most models have achieved good results in the ideal environment, but the performance of the interference model of environmental factors is significantly depressed in the actual detection. Given the limitations of traditional detection methods and the shortcomings of current deep learning detection algorithms, to improve the detection accuracy, detection speed, and the robustness of detection methods, this paper designs a new, efficient, and anti-interference bridge crack detection network based on the theoretical knowledge of deep learning.

The main contributions of this study are as follows:

- A deep separable deformation convolution operation is used to replace the conventional convolution operation, which optimizes the fitting ability of the prediction box and improves the feature extraction ability of the model.

- An inception module is introduced to expand the network width while controlling the increase in the number of parameters, which reduces the calculations required for detection, recognition, and classification and improves the detection speed of the model.

- The feature recalibration module, which combines the channel attention mechanism and spatial mechanism, is applied to improve the detection accuracy of the network by suppressing unimportant features while enhancing important features in space and channels.

The remainder of this paper is organized as follows. In the second section, we introduce related studies. In the third section, we introduce the details of the proposed method. In the fourth section, the details of the experiments on the proposed network and their results are presented. Finally, in fifth section, a brief conclusion is presented.

## 2. Preliminaries and related work

Currently, object detection is widely used for intelligent video surveillance, fault detection, medical treatment, and other fields. Many scholars have proposed different types of object detection models based on convolutional neural networks, which can be roughly divided into two types. The first type is a candidate region-based object detection model, represented by a regional convolutional neural network (R-CNN) and a real-time regional recommendation convolutional neural network (Faster R-CNN), which divides the detection process into two steps. In the first step, feature information is extracted from the input image according to candidate region selection algorithms (such as selection search [19] and edge search [20], etc.). The second step is to classify and adjust the position of the feature information obtained from the candidate region and finally output the object detection results. Although these models have high accuracy, the detection speed is slow, and it is difficult to meet the real-time requirements of bridge crack detection.

The second type is a regression-based object detection model represented by a single-shot multi-box detector (SSD) [21] and unified real-time object detection (YOLO) [22]. Compared to object detection models based on candidate regions, regression-based object detection models have a faster detection speed.

Fig 1 shows the structure of the SSD, which is divided into three parts: the main layer based on VGG16(very deep convolutional networks for large-scale image recognition) [23], the

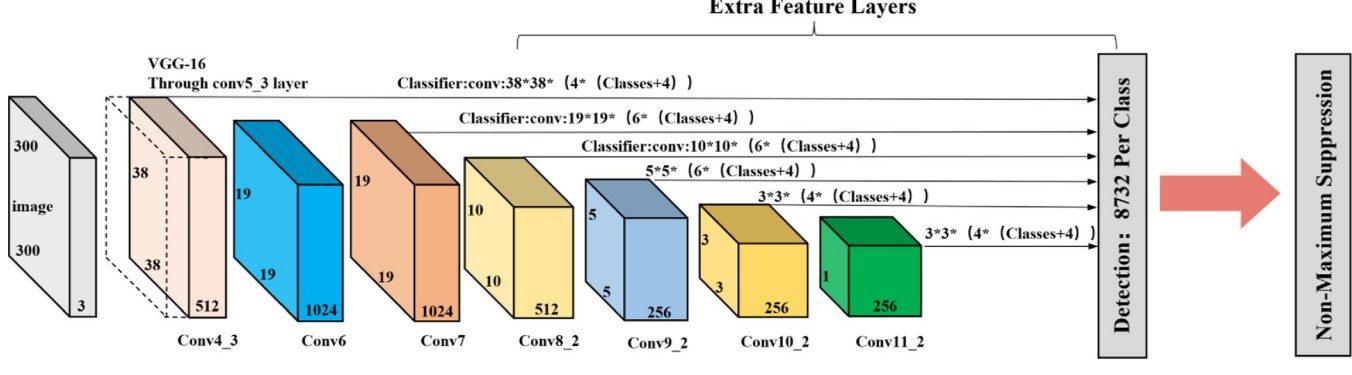

**Fig 1. Structure of SSD.**

feature extraction layer, and the classification layer. The VGG16 network structure in the main layer was optimized. First, the sixth and seventh convolution layers, Conv6 and Conv7, are used to replace FC6 and FC7 (fully connected layers) in the original structure of VGG16 to avoid the interference of the full connection layer with the detection object features and position information. Second, the feature mapping relationships of conv4_3, conv7_2, conv8_2, conv9_2, conv10_2, and conv11 are combined to form a multiscale feature extraction layer in the SSD. Finally, a 3×3 convolution is used to calculate the output feature graphs of the detection layer one by one to obtain the confidence required for detection target classification in the target detection task, and another 3×3 convolution is used to obtain the position information required for detection target regression in the object detection task. SSD adopts the method of multiscale object detection and applies an end-to-end learning model for bridge crack detection. Bridge cracks and pavement cracks are equal to concrete surface cracks. Based on the engineering application prospect of the crack detection method, this paper analyzes some representative crack detection networks based on SSD. Yan et al. [24] designed a pavement crack detection network based on SSD network by integrating the idea of deformation convolution, which improved the accuracy of network crack detection. Yang et al. [25] embedded the receptive field enhancement module into the SSD network to enhance its ability for crack feature extraction and improve the crack detection accuracy of the network. Feng et al. [26] also proposed an accurate bridge crack-detection algorithm based on an SSD. These algorithms can achieve good detection results under the conditions of a simple background and no interference but cannot meet the accuracy and speed requirements of bridge crack detection under complex conditions. Therefore, the original SSD model was optimized in this study to improve the detection accuracy and speed.

## 3. Methodology

The overall architecture of our proposed ISSD is shown in Fig 2, which introduces a composite structure with the DSDCM (Depth-Separable Deformation Convolution Module) and the IM (Inception Module) as the main components to extract features efficiently, and the FRM (Feature Recalibration Module) used to enhance the weight of effective features. Specially, given an image $x$ with size 300×300×3 and generate feature maps with sizes of 38×38×512, 19×19×1024, 10×10×512, 5×5×256, 3×3×256 and 1×1×256 respectively after different stages of DSDCM and composite structure processing. Then FRM is introduced to calibrate the characteristic relationship between channels and suppress interference information for the above characteristic maps of different scales to lay a solid foundation for the fusion of subsequent multi-scale characteristic maps. Finally, the fused characteristic map outputs the final detection results under the action of NMS (Non-Maximum Suppression). More implementation details regarding DSDCM, IM, and FRM are described in the following subsections.

### 3.1 Depth separable deformation convolution module (DSDCM)

**3.1.1 Deformation convolution.** Conventional convolution kernels are usually of fixed size (e.g., 3×3, 5×5, and 7×7), whereas the adaptability of the model to the geometric deformation of objects is almost entirely due to the diversity of the data [27]. In the bridge crack detection task, the conventional convolution has an insufficient fitting ability for narrow and long-strip bridge cracks, which leads to a low accuracy of the detection results. Therefore, we adopted the convolutional kernel distribution form, which offsets the position of each sampling point in the conventional convolution kernel to shift the position of the sampling point, realizing arbitrary deformation of the convolution, aiming to enhance the feature extraction

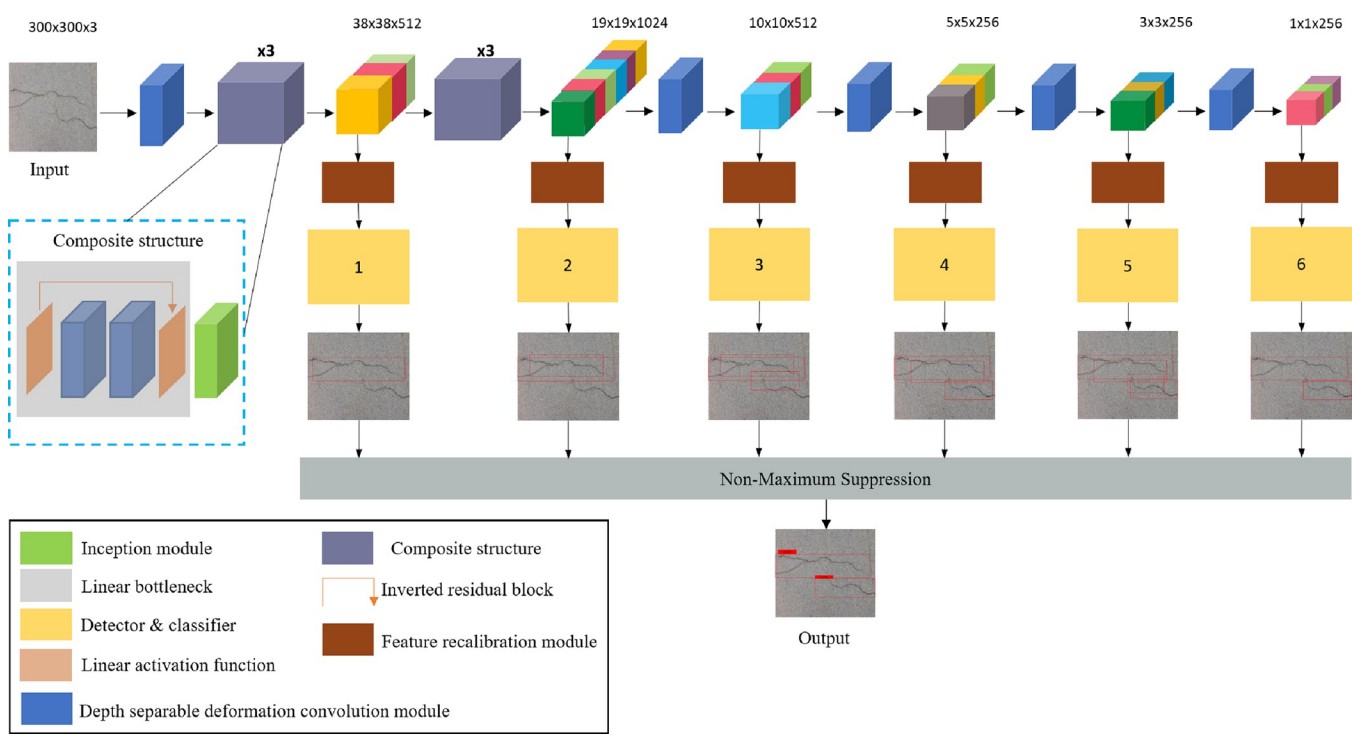

**Fig 2. Structure of ISSD.**

ability of the model for bridge cracks. Typical cases of deformable convolutions are shown in Fig 3.

The conventional convolution operation is divided into two steps [28]. In the first step, region R, which corresponds to the receptive field of the convolution kernel, is sampled on the input feature map. The second step is to successively sum the values of each sampling point and the weights of the corresponding convolution kernel positions. Region R defines the size of the receptive field, as shown in Eq 1.

$$R = \{(-1,\ 1),\ (-1,\ 0),\ \cdots(0,\ 1),\ (1,\ 1)\} \tag{1}$$

A point is convoluted in the output characteristic graph y:

$$y(p_0) = \sum_{p_n \in R} \omega(p_n) \bullet x(p_0 + p_n) \tag{2}$$

where $p_n$ represents the elements in the receptive field and $x$ is the input feature graph.

The deformed convolution kernel is obtained by shifting each element in the conventional convolution receptive field $R$.

$$y(p_0) = \sum_{p_n \in R} \omega(p_n) \bullet x(p_0 + p_n + \triangle p_n) \tag{3}$$

The offsets are $\{\triangle p_n | n = 1,2,3\cdots,N\}$ and $N$ is the number of elements in the receptive field $R$.

Because the offsets are not integers, deviations exist between the location of the sampling points and the actual pixel points of the feature map after the deformation operation of the

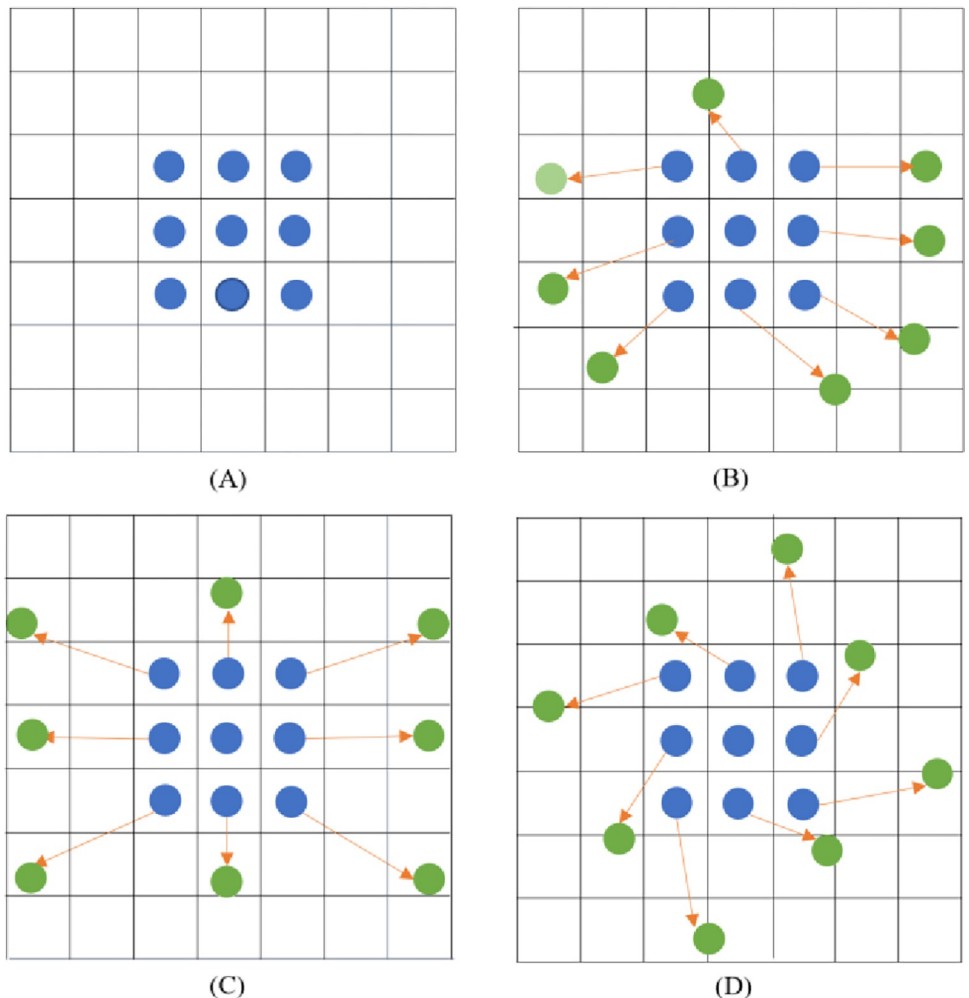

**Fig 3. Three typical cases of convolution kernel shifting of conventional convolution.** Group A is the conventional distribution, Group B is the distribution after arbitrary migration, Group C is the distribution after scaling transformation and Group D is the distribution after rotational transformation.

convolution kernel. In this case, a bilinear interpolation method was used for processing.

$$x(p) = \sum_q G(q,p) \bullet x(q), \tag{4}$$

$$x(p) = \sum_q g(q_x, p_x) \bullet g(q_y, p_y) \bullet x(q), \tag{5}$$

$$x(p) = \sum_q \max(0, 1 - |q_x - p_x|) \bullet \max(0, 1 - |q_y - p_y|) \bullet x(q), \tag{6}$$

where $p = (p_0 + p_n + \triangle p_n)$, $q$ enumerates the positions of all integral spaces in the feature graph, and $G(.,.)$ represents the bilinear interpolation kernel.

**3.1.2 Depth separable convolution.** Conventional convolution operations combine channel and dimensional mappings. Depth separable convolution [29] deals not only with the spatial dimension but also with the relationship between the depth dimension and channel.

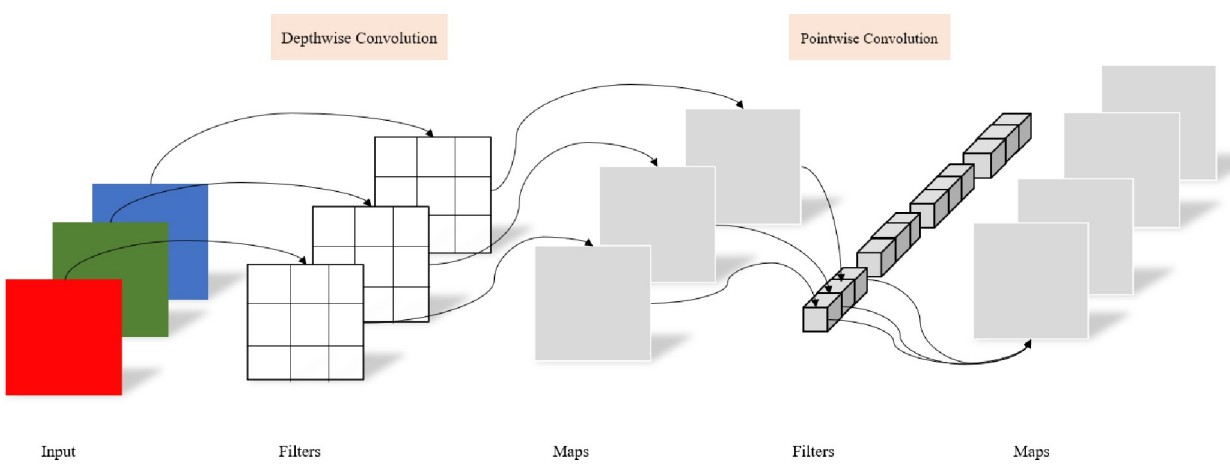

**Fig 4. Schematic diagram of depth separable convolution.**

Conventional convolution performs convolution operations on all channels in the input image areas. Depth separable convolution uses different convolution kernels on different channels to perform convolution operations. The operation processes of deep separable convolution are divided into two steps. The first step is a channel-by-channel convolution operation, in which three feature images are generated by the deep convolution operation. The second step is a point-by-point convolution operation, in which the three feature images generated by channel convolution are weighted in the depth direction, and a new feature map is generated. The deep-separable convolution processes are illustrated in Fig 4.

The number of parameters and amount of computation in the convolution affect the detection speed of the model. Compared with conventional convolution kernels, deep separable convolution kernels significantly reduce the number of parameters and the amount of computation.

Assuming that the input image is $D_k{\bullet}D_k{\bullet}M$, the size of the convolution kernel is $D_f{\bullet}D_f{\bullet}M$ and its number is $N$.

The number of parameters of conventional convolution is then calculated by

$$Z_r = D_k \bullet D_k \bullet D_f \bullet D_f \bullet M \bullet N \tag{7}$$

The number of parameters of deep convolution is calculated by

$$Z_d = D_k \bullet D_k \bullet D_f \bullet D_f \bullet M \tag{8}$$

The number of parameters of point-by-point convolution is calculated by

$$Z_p = D_k \bullet D_k \bullet M \bullet N \tag{9}$$

The number of parameters of deep separable convolution is calculated by

$$Z_D = D_k \bullet D_k \bullet M(D_f \bullet D_f + N) \tag{10}$$

By comparing Formula 7 with Formula 10, it can be seen that depth separable convolution can greatly reduce the number of model parameters and improve the efficiency of model operation by decoupling spatial and depth information.

**3.1.3 Depth separable deformable convolution.** A deformable convolutional network is a variant of a convolutional neural network that is very effective for solving complex visual

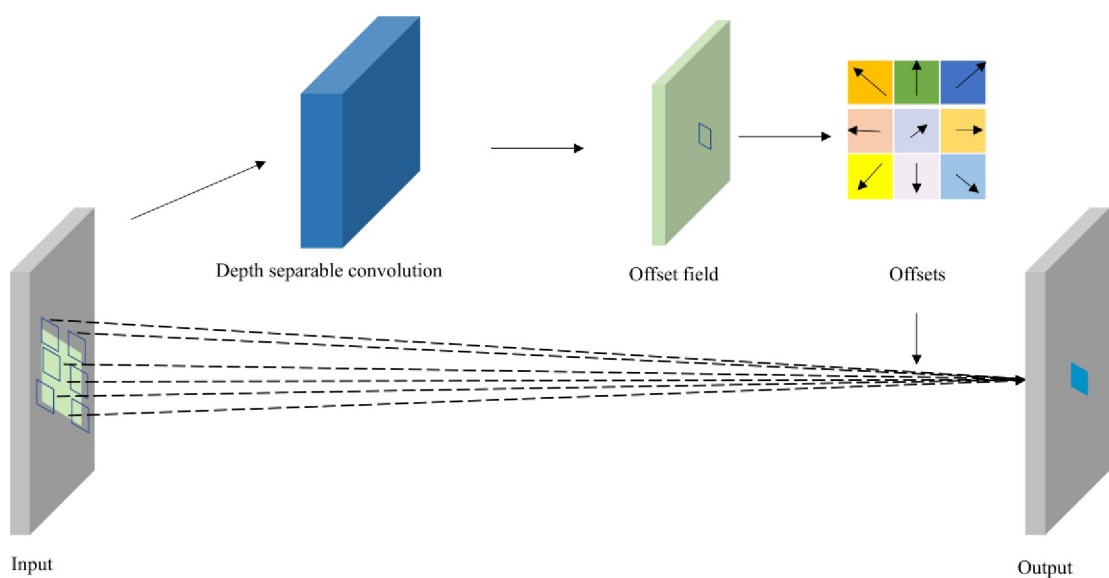

**Fig 5. Schematic diagram of the depth separable deformable convolution.**

tasks and learning dense spatial changes. Depth separable convolution mainly achieves model acceleration by decoupling spatial and depth information, which can further reduce the detection time of the model in practical applications. In this study, deep separable convolution is mainly integrated into the process of deformable convolution, and a deep separable deformable convolution module is proposed, which can further enhance the feature extraction ability, reduce the number of parameters, reduce the size, and improve the running speed of the network model. The entire process of deep-separation deformable convolution was divided into three steps, as shown in Fig 5. First, deep separation deformable convolution was used to sample the input feature maps to obtain the offset of each pixel point. Subsequently, a bilinear interpolation algorithm was used to obtain the pixel amount of each pixel point offset, which is equivalent to generating deformation on the convolution kernel to achieve the purpose of sampling the variable shape. Finally, the deformable convolution operation was performed with the convolution kernel with an offset on the input feature maps. Deep separation deformable convolution reduces the number of network parameters, and hence not only improves the speed of network operation but also improves the degree of network sparseness and enhances the ability of network feature extraction.

### 3.2 Inception module (IM)

The most direct way to improve the deep neural network is to increase the scale of the network. It includes increasing the depth and width of the network. Considering that the edge feature is the main feature of the cracks, the deepening of the network will lead to the decline of shallow feature learning ability, which is not conducive to detecting the cracks. Therefore, we choose to increase the size of filter banks in each layer, increase the width of the network, and then improve the network's performance. However, this method has two shortcomings: 1) A larger size usually means more parameters, and it is easier to cause overfitting of the network, especially in the case of insufficient samples. 2) Even if the size of each layer of the network is increased evenly, the total amount of computation will be increased sharply. Moreover, many operations will be wasted when the network capacity is underutilized. Inspired by reference [30], we adopted a sparse processing method in the feature dimension to alleviate the

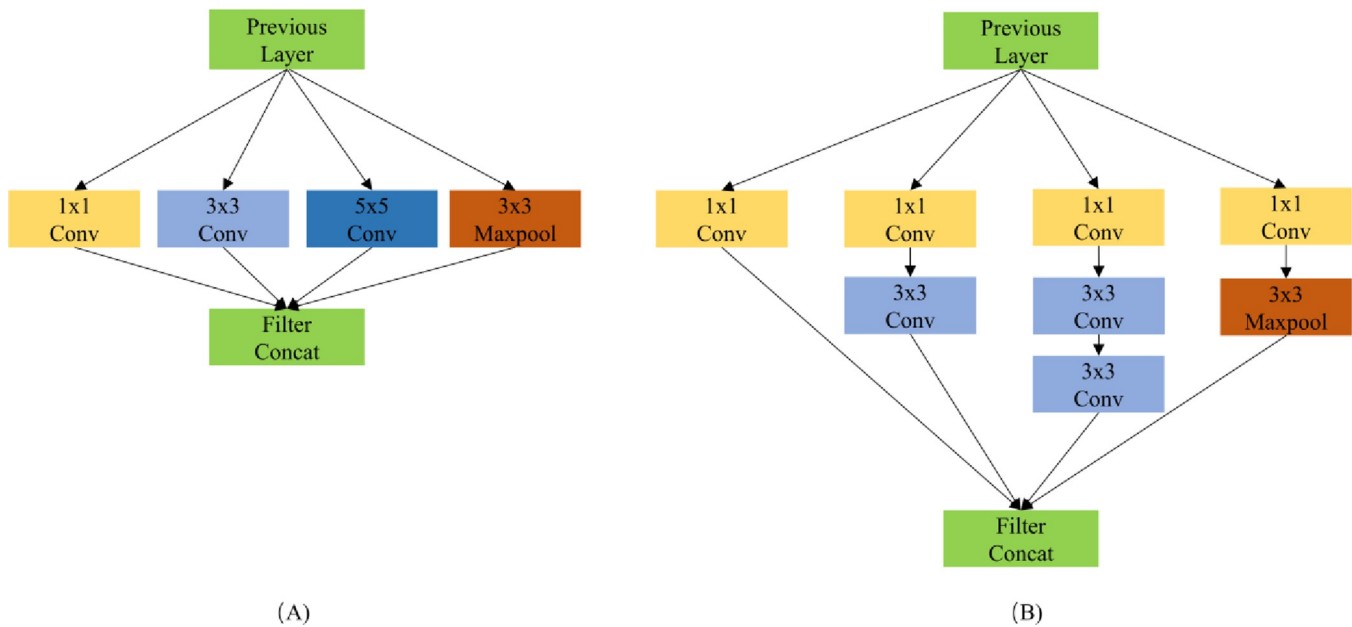

**Fig 6. Schematic diagram of the inception structure.** Group A represents the structure of the original inception and Group B represents the structure of the improved inception.

shortcomings caused by increasing the network width. Specially, an inception module in an SSD network was introduced to increase the network parallel computing ability and reduce the number of parameters. In addition, considering the limitation of computing resources of the field crack detection platform, we optimize it based on the original inception module. Specifically, we used 1×1 convolution as a reduced layer to reduce the number of channels and the amount of calculation. Further, we employed two 3×3 convolution to replace the 5×5 convolution in the original structure to achieve less parameter calculation under the premise of the same receptive field [31]. Compared with the original structure, the optimized structure not only reduces the number of module parameters, improves the reasoning speed of the whole network, but also increases the adjustment performance of the module to the dimension of the feature map, realizes cross-channel information combination, and is conducive to the enhancement of the overall detection performance of the network. The structure comparison of inception module is shown in Fig 6.

The introduction of inception modules in the feature extraction stage of the optimized model improved the feature fusion capabilities in the hidden layers of the model and fully broadened the channels of contextual information sharing. This method was helpful in improving the feature extraction efficiency of the bridge crack model. Although the optimized model increased the structural redundancy and the number of parameters, the changes in the parameters in the feature layers were controlled in a small range, and the feature extraction results were normalized in batches before entering the recognition layer. Thus, the increases in calculation were not obvious, and the detection speed of the model was improved.

### 3.3 Feature recalibration module (FRM)

To make the network pay more attention to channel features with effective information, suppress irrelevant features, and calibrate the feature relationship between channels, FRM was introduced into the network. The FRM adopted the design idea of squeeze-and-excitation

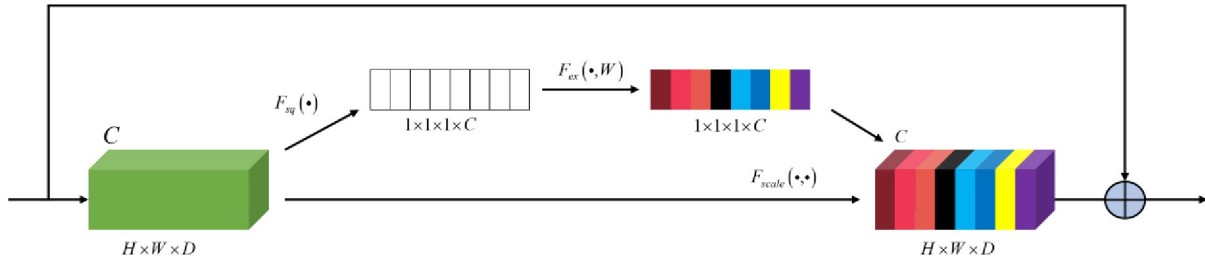

**Fig 7. Structure of the feature recalibration module.**

networks (SENet) [32], and its structure is shown in Fig 7. The FRM process can be divided into three steps. First, the input feature maps were compressed to obtain global information, and the specific formula is as follows:

$$z_c = F_{sq}(u_c) = \frac{1}{H \times W} \sum_{i=1}^{H} \sum_{j=1}^{W} u_c(i,j), \tag{11}$$

where $H$ and $W$ represent the height and width of the characteristic graph, respectively. $u_c$ represents the c-th channel of the characteristic graph; $u_c(i,j)$ represents the pixels of the i-th row and j-th column in the c-th channel; and $z_c$ represents the output after the compression operation.

Second, the channel features obtained by the compression operation are activated to generate the weight of each channel. The specific formula is as follows:

$$s = F_{ex}(z,w) = \sigma[w_2\varphi(w_{1z})], \tag{12}$$

where $w_1$ and $w_2$ represent the fully connected operation, $s$ represents the weight generated, $\varphi$ represents the ReLU activation function, and $\sigma$ represents the sigmoid function.

Finally, the weight generated in the activation operation is assigned to different channels, and the specific formula is as follows:

$$y_c = F_{scale}(u_c, s_c) = s_c \cdot u_c, \tag{13}$$

where $y_c$ represents the output matrix of the c-th channel.

FRM automatically obtained the importance of each feature channel through learning, improved the useful features according to their importance, and suppressed the features that were not useful for the current task; thus, the weight of the effective feature map was large, and the weight of the invalid or small effect feature map was small, which improved the detection of bridge cracks.

### 3.4 Loss function

The loss function of the entire network is composed primarily of position loss ($L_{loc}$) and confidence loss ($L_{conf}$). The position loss uses the *SmoothL*1 loss function to calculate the error between the ground truth box and the prediction box, and the confidence loss uses the *Soft*max loss function to calculate the correct detection probability. The total loss function of network includes classification loss function and regression loss function. The specific formula of the total loss function of the network is as follows:

$$L(x,c,l,g) = \frac{1}{N}\left[L_{conf}(x,c) + \alpha L_{loc}(x,l,g)\right], \tag{14}$$

where $L(x,c,l,g)$ denotes the total loss function, $c$ represents the degree of confidence, $l$ represents the prediction box, $g$ represents the ground truth box, $N$ represents the number of matches between the ground truth box and the prediction box, $\alpha$ represents the weight coefficient, and $x = \{0,1\}$. When the IOU (intersection over union) is greater than the threshold (set to 0.5 in this study), $x = 1$. Otherwise, $x = 0$.

The specific formula of the position loss function is as follows:

$$L_{loc}(x,l,g) = \sum_{j}^{N} \sum_{i\in P}^{N} \sum_{m\in\{o_x,o_y,w,h\}} x_{ij}^{k} SmoothL1(l_i^m - \widehat{g_j^m}), \tag{15}$$

$$SmoothL1 = \begin{cases} 0.5x^2 & |x| < 1 \\ |x| - 0.5 & |x| \geq 1 \end{cases}, \tag{16}$$

where $i\in P$ represents the i-th prediction box area as a positive sample, $o_x$ and $o_y$ represent the offsets in the $x$ and $y$ directions between the center of the prediction box or the ground truth box and the default box, respectively; $w$ and $h$ represent the deviation between the width and height of the prediction box or the ground truth box and the default box, respectively; $x_{ij}^k \in \{0,1\}$ (when the i-th prediction box matches the j-th ground truth box, $x_{ij} = 1$, otherwise $x_{ij} = 0$), $\widehat{g_j^m}$ is the position parameter of the ground truth box after encoding, and $l_i^m$ is the position parameter of the prediction box.

The specific formula of the confidence loss function is as follows:

$$L_{conf}(x,c) = -\sum_{j}^{N_1} \sum_{i\in P}^{N_1} x_{ij}^p \log(\hat{c}_i^p) - \sum_{i\in N_1} \log(\hat{c}_i^0), \tag{17}$$

$$\hat{c}_i^p = \exp(c_i^p)/\sum_{p} \exp(c_i^p), \tag{18}$$

where $i\in N_1$ represents the i-th prediction box area as a negative sample, $\hat{c}_i^0$ is the probability that the prediction box is the background, and $\hat{c}_i^p$ represents the probability calculated by the *Soft*max function.

## 4. Experiment and results

The bridge crack dataset used for model training and testing was introduced, and the effectiveness of each method was verified separately. The proposed network was then compared with FCN [33], SSD, U-Net [34], CrackDFANet [35], LDCC-Net [36], FPHBN [37], and (ABCNet) Network in reference [38]. Finally, conclusions were drawn by analyzing the experimental results.

### 4.1 Experimental dataset and computer environment

In this study, two crack datasets were used as samples: the SDNET [39] and CCIC datasets [40]. The images in the SDNET dataset were collected from walls, roads, and bridge surfaces. The entire dataset contained more than 56000 images which were divided into those with and without cracks. The CCIC dataset collected 40000 images of cracks and noncracks. The inherent data hunger of deep learning network makes the network training need massive data as support. This paper analyzes the characteristics of data samples in SDNET and CCIC, and constructs a new dataset according to the data fusion principle of similar label merging. To better

**Table 1. Number of different samples in the WCD dataset.**

| Type | SDNET | | CCIC | |
|---|---|---|---|---|
| | Crack | Non-crack | Crack | Non-crack |
| Train | 6000 | 6000 | 5000 | 5000 |
| Test | 2000 | 2000 | 2000 | 2000 |

**Table 2. Specific index parameters of the workstation.**

| Hardware/Software | Specification/Parameters/Version |
|---|---|
| CPU | Intel Core i5 8 Generation |
| GPU | NVIDIA GeForce GTX1060/6GB |
| RAM | 8GB |
| Anaconda | 3–5.1.0 |
| Python | 2.7.5 |
| TensorFlow | 1.10 |

train the network, this paper adopts 300×300 fixed size nonoverlapping clipping windows to randomly selected 16000 images from the SDNET dataset and 14000 images from CCIC to combine into a larger crack dataset called the WCD dataset. Samples from the WCD dataset were proportionally divided into a training set and a test set. The number of different samples in the WCD dataset is listed in Table 1.

The bridge crack-detection model proposed in this study is a program environment built in the TensorFlow framework. The experimental hardware was a Dell Precision T3630 workstation; the specific parameters of the workstation are listed in Table 2.

## 4.2 Evaluation indicators

There are various object detection models based on the convolutional neural network, and the principles of object detection are different for different detection models. To quantitatively evaluate the detection performance of the models, we must establish a corresponding reference standard to evaluate the detection performance of all models comprehensively and objectively. The commonly used evaluation indexes of object detection include the mean accuracy and number of images processed per second (FPS) [41].

The confusion matrix is the most basic and intuitive method for measuring an object detection model [42]. The confusion matrix includes the following four indicators: ① The true value is positive, and the model considers it to be positive (true positive = TP). ② The true value is positive, but the model considers it to be negative (false negative = FN). ③ The true value is negative, but the model considers it positive (false positive = FP). ④ The true value is negative, and the model considers it to be negative (true negative = TN). False negatives are statistical errors of the first type, and false positives are statistical errors of the second type.

Because the indicators in the confusion matrix count the number of samples, it is difficult to accurately evaluate the model using only the number of samples when processing a large amount of data. Four secondary indicators were extended from the basic statistical results of the confusion matrix: accuracy, accuracy rate, and recall rate.

Accuracy is the proportion of all correctly judged results in the model to the total observed values, and its formula is shown in Eq (18).

$$Acc = \frac{(TP + TN)}{(TP + TN + FP + FN)} \qquad (19)$$

Accuracy rate refers to the proportion of all results in which the model prediction is positive and correct. Its formula is shown in (19).

$$\Pr e = \frac{(TP)}{(TP + FP)} \tag{20}$$

The recall rate refers to the proportion of all results whose true values are positive and correctly predicted by the model. Its formula is shown in (20).

$$\Re c = \frac{(TP)}{(TP + FN)} \tag{21}$$

Since precision and recall are a pair of contradictory indicators, in general, when the precision value is high the recall value is often low, and when the recall value is high the precision value is often low. In order to comprehensively consider the influence of these two indicators, *F−measure* (weighted harmonic mean of Precision and Recall) is proposed, and its formula is expressed as shown in (21).

$$F - measure = 2\frac{(\Pr e \times \Re c)}{(\Pr e + \Re c)} \tag{22}$$

*F−measure* not only improves the precision and recall rates but also ensures that the gap between them is narrowed as much as possible to measure the detection efficiency of the model more comprehensively.

In the statistical analysis of the model test results, the recall rate value is typically used as the abscissa, and the precision rate value is used as the ordinate to draw the P–R curve. By observing the fluctuation of the P–R curve, the precision rate can be negatively correlated with the recall rate value. IOU [43] reflects the correlation between the predicted value detected by the model and the real value of the objects. IOU was calculated as follows:

$$IOU = \frac{area(B_{\mathrm{det}} \cap B_{gt})}{area(B_{\mathrm{det}} \cup B_{gt})}, \tag{23}$$

where $B_{det}$ represents the size of the detection box, $B_{gt}$ represents the size of the calibration box of the detection target, $area(B_{det} \cap B_{gt})$ represents the coincidence area of the two boxes, and $area(B_{det} \cup B_{gt})$ represents the total area of the two boxes combined.

The higher the correlation, the higher is the IOU. In the model training process, thresholds of different IOU should be set to measure the detection accuracy of the model. Experimental results in [36] show that it is appropriate to set the threshold value of IOU as 0.5 in the bridge crack detection task. The accuracy of model detection is usually described by a precision–recall curve (PR curve). The PR curve takes the recall rate as the vertical axis and accuracy as the horizontal axis. The accuracy curve of the recall rate is commonly used to measure the detection performance of the models. The curve generally showed that the recall rate was low when the accuracy was high, and when the recall rate was high, the accuracy was low.

## 4.3 Network training

Learning rate is a key parameter in network training. An unreasonable learning rate will lead to the problem of gradient explosion or gradient disappearance of the network and failure to complete the training. A reasonable learning rate will promote network convergence. The relationship between the loss function value and number of epochs at different learning rates in the network training process is shown in Fig 8. The curve variation trend indicated that when the learning rate was 0.0001, the loss function curve declined slowly, and a long time was

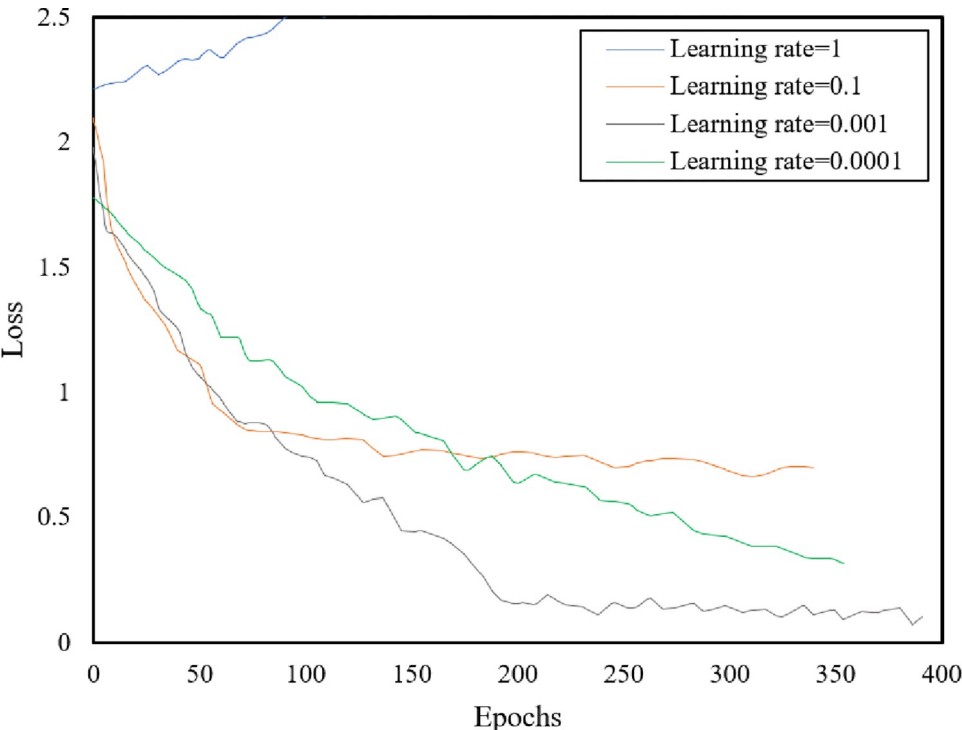

**Fig 8. Curves of training loss with different learning epochs.**

required to reach convergence. When the learning rate was 0.001, the loss function curve decreased rapidly and converged within a short time. When the learning rate was 0.1, the loss function curve decreased rapidly in the early stages and gradually in the later stages. When the learning rate was 1.0, a gradient explosion occurred in the training stage of the network, and the network could not complete the training. Therefore, the learning rate was set as 0.001. Furthermore, this paper compares three commonly used gradient descent methods, namely batch gradient descent method (BGD), random gradient descent method (SGD), and small-batch gradient descent method (MBGD). BGD sacrifices speed while pursuing accuracy. Too slow convergence speed can not meet the timeliness requirements of detection. On the contrary, SGD adopts the strategy of reducing iterative samples to improve the update speed of each round of parameters. However, it isn't easy to ensure detection accuracy. Considering the small scale of bridge crack samples and the large sample size of the data set, we use MBGD with a batch size of 32 and weight attenuation of 0.0001 to balance speed and accuracy.

Epoch refers to sending all training samples to the network to complete forwarding calculation and backpropagation. With the increase in the number of epochs, the number of weight update iterations increases, and the network's performance also changes. A reasonable number of iterations is the key to practical training the network to achieve the best state. Fig 9 shows the results under the different epochs. When the number of iterations is small (100 epochs), the network is in the state of fitting, and the detection effect is poor, resulting in the loss of 6 parts in the results. As the number of iterations increases to 160 epochs, the detection performance of the network is gradually improved, the detection accuracy of the network is improved, and the missing detection part is reduced to 4. When the number of iterations reaches about 220, the detection performance of the network comes the best. However, due to the crack scale, there is still a lack of some detailed features. When the number of iterations is

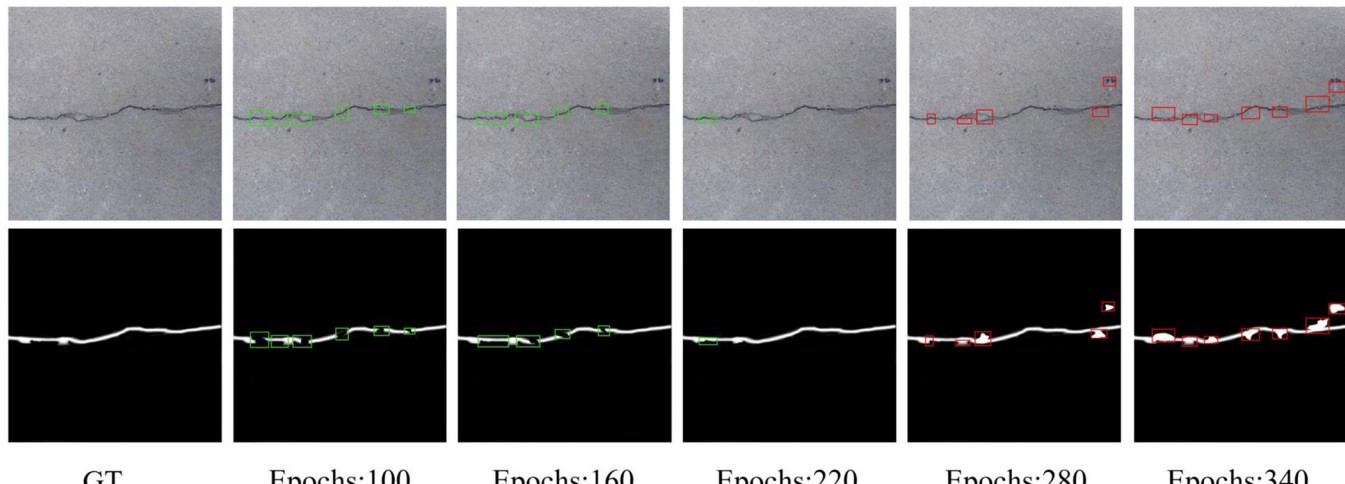

**Fig 9. Comparison of output results at different epochs.** The green boxes locate the missing detection parts of the detection results and the red boxes locate the false detection parts of the detection results.

280 epochs, the number of detected errors will be increased from the area of no cracks to the size of no cracks. This shows that the network has the trend of overfitting. With the increase of training times to 340 times, the network presents the state of overfitting, the area of false detection object increases to 7, and the area of incorrect detection area will not expand. Therefore, we set the number of iterations to 220, maintaining the convergence consistency of network training loss under different learning rates.

In addition, other specific training parameters were set during the training process, as listed in Table 3.

### 4.4 Effectiveness of network structure

To evaluate the effectiveness of the components of the proposed ISSD, ablation experiments were performed on the WCD dataset. SSD(VGG-16) was used for comparison with the same parameter settings, and the experimental results are presented in Table 4.

Several conclusions can be drawn from the results. First, the proposed network ISSD can achieve superior performance compared to other networks (Precision: 0.9053, Re: 0.9116, F-measure: 0.9084, FPS: 75). Second, the components of the proposed ISSD can improve the detection accuracy and speed of the network. Comparing the results of SSD with and without the proposed components, it can be seen that the DSDCM improved the F-measure by 4%, the IM enhanced the FPS of the network by 21%, and the FRM increased the F-measure by 5%. Third, different components optimize the network to different degrees. In particular,

**Table 3. Network parameter setting.**

| Type | Parameter | Value |
|---|---|---|
| training | Initial learning rate | 0.001 |
| | Momentum | 0.9 |
| | Weight decay | 0.0001 |
| testing | Initial learning rate | 0.001 |
| | Momentum | 0.9 |
| | Weight decay | 0.0001 |

**Table 4. Validation results of components in the ISSD.**

| Network | Accuracy | Precision | Recall | F-measure | FPS |
|---|---|---|---|---|---|
| SSD | 0.7835 | 0.7795 | 0.7846 | 0.7820 | 53 |
| SSD+DSDCM | 0.8321 | 0.8249 | 0.8297 | 0.8273 | 52 |
| SSD+IM | 0.8153 | 0.8048 | 0.8131 | 0.8089 | 60 |
| SSD+FRM | 0.8415 | 0.8357 | 0.8386 | 0.8371 | 54 |
| ISSD | 0.9153 | 0.9053 | 0.9116 | 0.9084 | 75 |

comparing the results of SSD + FRM and SSD + DSDCM, it can be seen that the accuracy of the former has a 1% advantage over the latter, which shows that FRM reduces the impact of negative samples on network accuracy and is conducive to the improvement of network accuracy. Comparing the results of SSD + DSDCM and SSD + IM, it can be seen that the detection speed of the latter is 60 FPS, which is higher than that of the former at 52FPS. The superior performance is due to IM improving the network parallel computing ability and reducing the number of parameters. Based on the comprehensive analysis of the above experimental results, the bridge crack detection network ISSD designed in this paper has dramatically improved the detection accuracy and detection speed compared with the original network (the accuracy advantage is about 13%, and the speed advantage is about 28%). The superior performance shows that the SSD algorithm has enough room for improvement in solving practical engineering problems and effectively promotes the application of SSD algorithm-based detection networks in the field of bridge cracks.

## 4.5 Comparison with state-of-the-art bridge crack detection networks

**4.5.1 Overall performance analysis.** To prove that the ISSD was more competitive than the other bridge crack networks, the proposed network was compared with the FCN, SSD, U-Net, CrackDFANet, LDCC-Net, FPHBN and ABCNet. All the networks were trained and tested on the same hardware platform using the same dataset. Table 5 shows a series of quantitative experimental results. On the whole, the F-measure of all networks is more than 0.8. which indicates that all networks have certain detection performance. Specifically, the F-measure of FCN is lower than 0.8, the F-measure of FPHBN and ABCNet are close to 0.87, and that of the remaining networks is about 0.89. The F-measure of ISSD is the most prominent, reaching 0.912, which shows that ISSD is good at capturing local details, which are often rich in texture features, and are very important in bridge crack detection.

Further, we compared the computational efficiency and computational complexity of all networks, as shown in Table 6. In the experiment, all networks run the same number of iterations under the same hardware platform and experimental settings. The results show that in all models, the floating-point computation of ISSD and LDCC is the lowest, far lower than that of

**Table 5. The results of different networks on the WCD dataset.**

| Network | Precision | Recall | F-measure |
|---|---|---|---|
| FCN | 0.801 | 0.791 | 0.796 |
| U-Net | 0.887 | 0.873 | 0.880 |
| CrackDFANet | 0.895 | 0.881 | 0.888 |
| LDCC-Net | 0.896 | 0.883 | 0.889 |
| FPHBN | 0.851 | 0.849 | 0.850 |
| ABCNet | 0.869 | 0.857 | 0.863 |
| ISSD | 0.901 | 0.917 | 0.912 |

**Table 6. The computational efficiency and computational complexity of different networks.**

| Network | Epochs | Flops | FPS |
|---------|--------|-------|-----|
| FCN | 220 | 15.4G | 28 |
| U-Net | 220 | 5.21G | 31 |
| CrackDFANet | 220 | 1.32G | 73 |
| LDCC-Net | 220 | 1.29G | 75 |
| FPHBN | 220 | 2.73G | 58 |
| ABCNet | 220 | 1.36G | 62 |
| ISSD | 220 | 1.28G | 77 |

other networks. In addition, ISSD and LDCC net's reasoning speed is outstanding, exceeding 73 FPS. Although LDCC is close to ISSD in terms of computational efficiency and computational resource complexity, based on Table 5, ISSD achieves the best detection effect with the highest computational efficiency and the lowest computational resource complexity.

To intuitively analyze and compare the detection performance of the network, we selected four kinds of crack samples for depth visual feature analysis. Fig 10 shows the detection results of the networks: single shape sample (row 1), composite shape sample (row 2), regional interference sample (row 3), and robust interference sample (row 4). The single sample detection results show that the all networks can describe the crack shape and its area to varying degrees., except for the interference of environmental noise. Specifically, the results of FCN, FPHBN, and ABCNet are disturbed by the environment to varying degrees. LDCC-Net and ISSD are more refined to extract crack texture features, which is conducive to detecting cracks. By analyzing the detection results of composite shape samples, it is found that the FCN network lags behind other networks in the expression of crack detail information, which is due to the complete convolutional structure of FCN. The detailed information is diluted in the progressive pooling operation, which affects the expression of local features of the network. The robust feature extraction ability and excellent negative sample screening ability of the network ISSD designed in this paper support its accurate expression of the crack shape of the composite

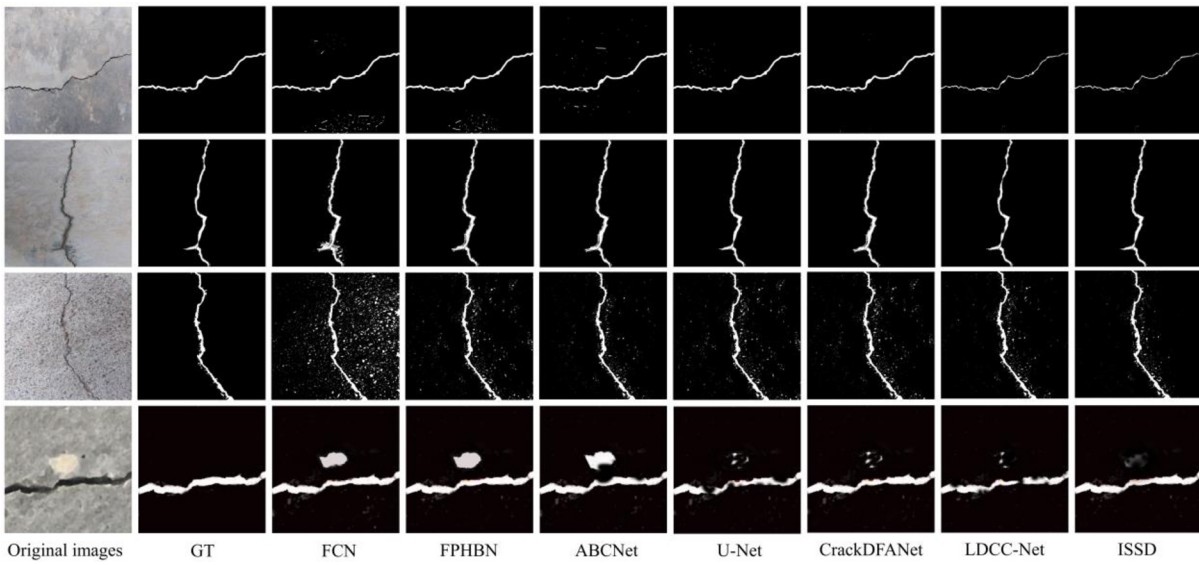

**Fig 10. Visualization of detection results of compared networks on the WCD dataset.**

structure. The background of the other two types of samples is complex. There are different degrees of interference. Compared with the detection performances of the first two samples, the performance of FCN is weakened to the greatest extent, and other networks are also significantly suppressed. U-net, CrackDFANet, and LDCC-Net cannot even wholly depict the shape of cracks. Such results show that for the network based on visual feature detection, the interference factors in the environment have a significant limit on the network performance. The ISSD network designed in this paper only achieves relatively stable detection results and cannot eliminate this limit.

**4.5.2 Performance in real images.** To further compare the anti-jamming capability of the network, we select four kinds of crack samples for depth visual feature analysis. Fig 11 shows the detection results of the network: the transverse crack sample under substantial interference (row 1), the transverse crack sample under shadow interference (row 2), the cross-crack sample under large-area interference (row 3), and mesh crack sample under substantial interference (line 4). The first set of experiments (row 1) shows substantial interference, which undoubtedly poses a challenge to crack detection. Although ISSD improves the noise reduction ability to a certain extent, it is still not ideal. For the input image of the shadow interference area (row 2), the detection results of all networks interfere to varying degrees. FCN and FPHBN have seriously interfered. The rest of the networks can alleviate the interference of shadow on the detection results to a certain extent, but ISSD can extract the crack texture features more fully, which shows that ISSD still maintains a robust feature ability under certain interference conditions. For the input image with extensive area interference (row 3), the optimization strategy adopted by the network is challenging to deal with due to the prominent visual characteristics of the background, and the detection results of the all networks are disturbed. Unlike the last three experiments, the crack shape of the samples selected in the fourth group is more complex, posing a new challenge to the network(row4). Due to the influence of background interference, the ability of the network to extract network crack features is reduced. In addition, the networks can not accurately predict the crack part's shape, and the detection effect is not ideal. From the processing results of the above four complex samples, the network proposed in this paper has achieved relatively stable optimization results in

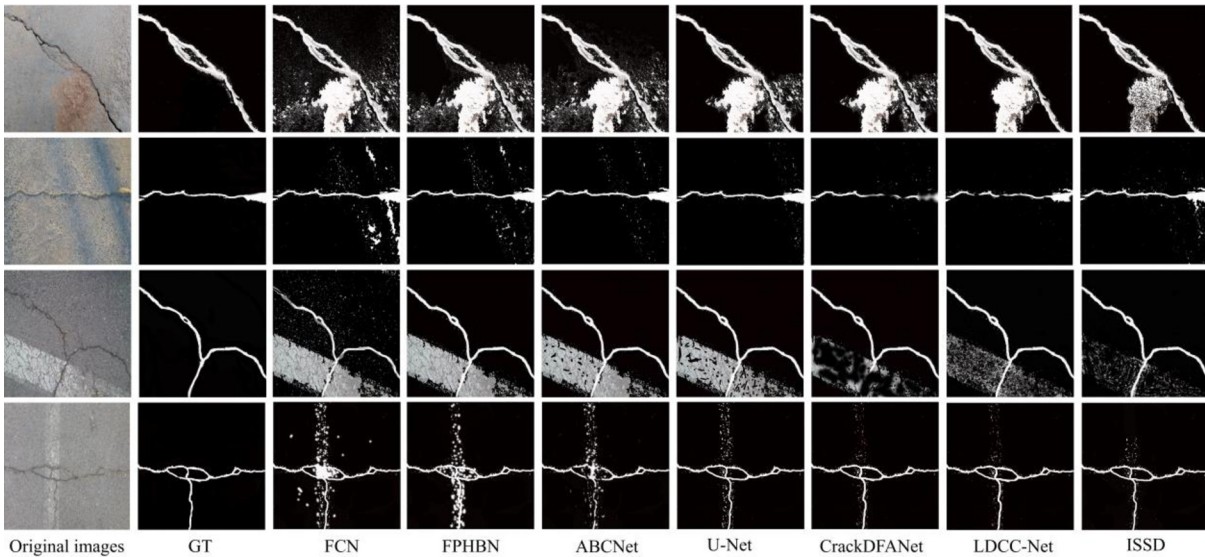

**Fig 11. The visualization of detection results of compared networks in real images.**

enhancing the target and reducing the interference. However, in terms of engineering practice, it is a severe challenge to realize the ability of anti-interference and anti-noise.

## 5. Conclusion

This study proposes a bridge crack detection model ISSD, which combines DSDCM, IM, and FRM closely and seamlessly. Specifically, DSDCM improves the crack feature extraction ability of the model, IM improves the reasoning speed of the network, and FRM alleviates the interference of irrelevant channel features. Further, a series of experiments show that compared with several existing crack detection networks, ISSD has better performance, reaching 0.912 F-measure and 77 FPS. Although the proposed ISSD method can obtain more satisfactory performance than other methods, the complexity of neural network structure and computing power requirements are significant challenges for the current portable bridge crack detection terminal. In addition, the anti-interference ability of the network is still difficult to overcome all kinds of environmental noise in engineering applications. We will focus on these issues in future research.

## Author Contributions

**Data curation:** Faming Shao.

**Methodology:** Qiang Wang.

**Project administration:** Xiaohui He.

**Software:** Jinkang Wang, Qunyan Jiang.

**Writing – original draft:** Guanlin Lu.

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
