## [Decision Letter · Decision Letter 0]

26 Apr 2022

PONE-D-22-07252Research on Bridge Crack Detection Based on Improved Single Shot Multi-Box DetectorPLOS ONE

Dear Dr. He,

Thank you for submitting your manuscript to PLOS ONE. After careful consideration, we feel that it has merit but does not fully meet PLOS ONE’s publication criteria as it currently stands. Therefore, we invite you to submit a revised version of the manuscript that addresses the points raised during the review process.

We look forward to receiving your revised manuscript.

Kind regards,

Ardashir Mohammadzadeh, Phd

Academic Editor

PLOS ONE

Journal Requirements:

Reviewers' comments:

Reviewer's Responses to Questions

**Comments to the Author**

1. Is the manuscript technically sound, and do the data support the conclusions?

Reviewer #1: Yes

Reviewer #2: Partly

2. Has the statistical analysis been performed appropriately and rigorously? 

Reviewer #1: Yes

Reviewer #2: N/A

3. Have the authors made all data underlying the findings in their manuscript fully available?

Reviewer #1: No

Reviewer #2: No

4. Is the manuscript presented in an intelligible fashion and written in standard English?

Reviewer #1: No

Reviewer #2: No

5. Review Comments to the Author

Reviewer #1: “An improved single-shot multi-box detector (SSD) called ISSD is proposed, which seamlessly

8 combines the depth separable deformation convolution module (DSDCM), inception module (IM), and feature 9 recalibration module (FRM) in a tightly coupled manner to tackle the challenges of bridge crack detection”. I have the following points which need to be properly answered.

Comment #1: The authors do not discuss the figures presented in the experimental results section. This is a critical issue in this section.‎

Comment #2: There are lots of typos. English needs to revise again with a professional editing service. Also, the figures are not clear in some cases.

Comment #3: Mention the limitations and future works of the developed system elaborately.

Comment #4: Several techniques have been described in the Introduction section. How do the authors outperform each of these reviewed systems? A clear statement is needed to highlight the contribution (use the table to discuss each method).

Comment #5: Discuss the stability of the system in terms of complexity.

Comment#6: Detailed evaluation of how the proposed algorithm performs and the varying parameters must be provided. You only provided a learning rate.

Comment#7: All the figures must be in 300dpi.

Comment#8: Parametric settings need to discuss. How you select the parameters in your model?

Comment#9: The conclusion is confusing. You need to change it. Should be according to the body of the manuscript. You must write some findings in the conclusion section.

Comment#10: All the figures are not cited properly in the text.

Reviewer #2: Comments:

1. In line 52, what subjective factors are in consideration should be indicated and elaborated.

2. In lines 59, 61, and 64, the complex environmental noise should be elaborated in terms of its nature and in terms of its effect on detection.

3. There are inconsistencies between SSD layer names in lines 95-100 and SSD layer names in Fig 1. In [21], there is a Conv6 layer as also indicated in line 96, however Figure 1 does not have it.

4. In general, Fig 2 is not digestible as a whole. Please give data structure information (i.e., dimensions) about colored boxes. There are color inconsistencies between figure legend and the figure. It is not evident where the composite structure sits in the architecture nor there is a reference about it in the text. Please carefully review the figure to make it easy to comprehend and more connected to the text.

5. References and the text are not consistent:

- [24] is not work of Zhao et al.

- Lu et al.'s work [25] is not about bridge crack detection as asserted by the authors.

- Feng et al.'s wok [26] is not about bridge crack detection. However, on can stress that they are correlated problems. Just align the previous work and your statements.

6. [28] is not about deformable convolutions. Please update all the references in the manuscript and make them consistent to the text. The manuscript is not scrutable with that state.

7. In Section 3.1.1, deformation convolution is illustrated however it does not give any credit to the its original inventors: https://arxiv.org/pdf/1703.06211.pdf Please cite properly. ([27] is not that paper.)

8. In Section 3.1.2, in line 160, the reference [29] is not about depth separable convolution. The DOI in [29], doi:10.4271/2014-01-0975, is not a paper titled Xception. It is titled as Fatigue Behavior of Stainless Steel Sheet Specimens at Extremely High Temperatures.

9. WCD dataset does not have any reference, thus not accessible.

NOTE: This manuscript has too many fundamental errors. Please carefully review the paper and then I will be able to read it with a mind in peace.

Thank you.

6. PLOS authors have the option to publish the peer review history of their article (what does this mean?). If published, this will include your full peer review and any attached files.

Reviewer #1: No

Reviewer #2: No

---

## [Author Response · Author response to Decision Letter 0]

23 May 2022

Original Manuscript ID: PONE-D-22-07252 

Original Article Title: Research on Bridge Crack Detection Based on Improved Single Shot Multi-Box Detector

To: PLOS ONE Editor

Re: Response to reviewers

Dear Editor,

Thank you for allowing a resubmission of our manuscript, with an opportunity to address the reviewers’ comments.

We are uploading (a) our point-by-point response to the comments (below) (response to reviewers), (b) an updated manuscript with yellow highlighting indicating changes (Supplementary Material for Review), and (c) a clean updated manuscript without highlights (Main Manuscript).

Best regards,

<Xiaohui He> et al.

Reviewer#1, Concern # 1: The authors do not discuss the figures presented in the experimental results section. This is a critical issue in this section.‎

Author response: We gratefully appreciate for your valuable suggestions. We re-combed the writing ideas of the experimental part, focused on the specific experimental results of different networks, and analyzed the deep-seated reasons behind the various experimental results from the level of target detection mechanism. The reviewers put forward valuable opinions, which increased the article's expression logic and helped improve the overall quality of the article.

Author action: We revised this part of the article and highlighted it. (Line433-444 and Line452-506).________________________________________

Reviewer#1, Concern # 2: There are lots of typos. English needs to revise again with a professional editing service. Also, the figures are not clear in some cases.

Author response: Thank you again for your positive comments and valuable suggestions to improve the quality of our manuscript. In view of these problems, we have updated the illustrations in the article. Unfortunately, our article has been polished by a commercial translation agency before submission, and we have further optimized the language part of the article. Although we are not very confident, we still hope it can meet your requirements. The English editing service certificate is as follows:

. 

Author action: We improved the resolution of the pictures in the article to 300dpi. The modified pictures are as follows:

FIG 1. Structure of SSD.

FIG 2. Structure of ISSD.

FIG 3. Three typical cases of convolution kernel shifting of conventional convolution. Group A is the conventional distribution, Group B is the distribution after arbitrary migration, Group C is the distribution after scaling transformation and Group D is the distribution after rotational transformation.

FIG 4. Schematic diagram of depth separable convolution.

FIG 5. Schematic diagram of the inception structure. Group A represents the structure of the original inception and Group B represents the structure of the improved inception.

FIG 6. Structure of the Feature recalibration module.

FIG 7. Curves of training loss with different learning epochs.

FIG 8. Comparison of output results at different epochs. The green boxes locate the missing detection parts of the detection results and the red boxes locate the false detection parts of the detection results.

FIG 9. Visualization of detection results of compared networks on the WCD dataset. From left to right: original images, GT, FCN, U-Net, CrackDAFNet, ISSD.

FIG 10. The visualization of detection results of compared networks on special cases. From left to right: original images, GT, FCN, U-Net, CrackDFANet, ISSD.

FIG 11. Precision-recall curves of composed networks on WCD dataset.

Reviewer#1, Concern # 3: Mention the limitations and future works of the developed system elaborately.

Author response: We gratefully appreciate for your valuable suggestions. In the conclusion of the article, we summarize the advantages and disadvantages of the network proposed in this paper, and formulate the direction of the next work.

Author action: We updated the conclusion of the article to make the structure of the full text more complete. (Line 518-526).________________________________________

Reviewer#1, Concern # 4: Several techniques have been described in the Introduction section. How do the authors outperform each of these reviewed systems? A clear statement is needed to highlight the contribution (use the table to discuss each method).

Author response: Special thanks to you for your good comments. This paper designs a bridge crack detection network based on the deep learning theory. The work of this paper is only a part of the bridge crack detection project. The advantage of the detection network based on computer vision is its learnability. The network can learn the relevant knowledge of crack detection from the data samples and can detect the bridge crack accurately and quickly by relying on the mighty computing power of the convolution neural network. Compared with other traditional detection technologies, crack detection methods mainly rely on artificial vision inspection or instrument signal characteristic analysis. The detection accuracy and speed lag behind the detection network designed in this paper. However, bridge crack detection is a complex engineering project. The whole project includes many hardware, software, and corresponding manual operations. The design of bridge crack in this paper is a systematic engineering project, so it is impossible to compare with other detection technologies from the overall perspective of bridge crack detection engineering.

Author action: We revised the description of highlighting the traditional detection methods and the advantages and disadvantages of the network designed in this paper. (Line39-41, Line51-54 and Line 62-82).________________________________________

Reviewer#1, Concern # 5: Discuss the stability of the system in terms of complexity.

Author response: Thanks again to the reviewer, we are also aware of this problem. Network parameter quantity is an index to evaluate the complexity of a network system based on deep learning. We add this part to the article and combine it with the network's accuracy and reasoning speed to analyze the network's stability from the perspective of complexity. In addition to the content shown in the manuscript itself, we tested it on different training and test sets and different computing platforms, and the test results did not change significantly. Therefore, our network has stable performance under low computing power dependence after sufficient training. Due to the article's length, we don't think putting these details in the manuscript is necessary.

Author action: We added this part in the manuscript. (Line 452-464 and Table 5).________________________________________

Reviewer#1, Concern # 6: Detailed evaluation of how the proposed algorithm performs and the varying parameters must be provided. You only provided a learning rate.

Author response: Special thanks to you for your good comments. Network parameters and training mode play a vital role in the final performance of the network. Combined with the characteristics of bridge crack detection, we show more detailed information on the network training process in the article and comprehensively analyze the network's performance in different gradient descent modes and different iterative stages. 

Author action: We show more details of the network training process in the article, analyze the speed and performance of network training under different gradient descent modes, and compare the network's performance under different iterative stages on this basis. (Line386-412 and Fig.8).________________________________________

Reviewer#1, Concern # 7: All the figures must be in 300dpi.

Author response: Thank you again for your positive comments and valuable suggestions to improve the quality of our manuscript. In view of these problems, we have updated the figures in the article.

Author action: The fonts in our picture has been replaced, and the resolution of the picture has been increased to 300ppi. ________________________________________

Reviewer#1, Concern # 8: Parametric settings need to discuss. How you select the parameters in your model?

Author response: Special thanks to you for your good comments. This question has been partially answered in question 6.

Author action: We show more details of the network training process in the article, analyze the speed and performance of network training under different gradient descent modes, and compare the network's performance under different iterative stages on this basis. (Line386-412 and Fig.8).________________________________________

Reviewer#1, Concern #9: The conclusion is confusing. You need to change it. Should be according to the body of the manuscript. You must write some findings in the conclusion section.

Author response: It is true as Reviewer suggested that conclusion in the article is not clear in expression. We rewrote the conclusion chapter, focusing on the overview of the work done and the results achieved in this paper. In addition, according to the problems found in the experimental process, we analyzed the shortcomings of the work done in this paper and made a direction for future work.

Author action: We updated the manuscript by adding the description of this part in the article. (Line 518-526).________________________________________

Reviewer#1, Concern # 10: All the figures are not cited properly in the text.

Author response: Thank you again for your positive comments and valuable suggestions to improve the quality of our manuscript. Before submitting the manuscript, we have arranged the article in strict accordance with the requirements of the journal. We will continue to communicate with the editor to determine the citation format of the pictures in the article.________________________________________

Reviewer#2, Concern # 1: In line 52, what subjective factors are in consideration should be indicated and elaborated.

Author response: Special thanks to you for your good comments. The core problem of the effect of the detection method based on manual design features is the quality of manual design features, which are usually completed manually, and subjective factors are inevitably introduced in the feature design process.

Author action: We updated the manuscript by modifying the description of this part. (Line 51-54)________________________________________

Reviewer#2, Concern # 2: In lines 59, 61, and 64, the complex environmental noise should be elaborated in terms of its nature and in terms of its effect on detection.

Author response: Thank you again for your positive comments and valuable suggestions to improve the quality of our manuscript. In view of these problems, we have updated the illustrations in the article. 

Author action: We updated the manuscript by adding the description of this part. (Line62-68)________________________________________

Reviewer#2, Concern # 3: There are inconsistencies between SSD layer names in lines 95-100 and SSD layer names in Fig 1. In [21], there is a Conv6 layer as also indicated in line 96, however Figure 1 does not have it.

Author response: Thanks again to the reviewer, we are also aware of this problem. We updated the content of Figure 1 to conform to the description of this part in the article.

Author action: We have updated the content of Figure 1. The modified Fig. 1 is as follows.

original image

Updated image________________________________________

Reviewer#2, Concern # 4: In general, Fig 2 is not digestible as a whole. Please give data structure information (i.e., dimensions) about colored boxes. There are color inconsistencies between figure legend and the figure. It is not evident where the composite structure sits in the architecture nor there is a reference about it in the text. Please carefully review the figure to make it easy to comprehend and more connected to the text.

Author response: Special thanks to you for your good comments. We have replaced Fig2. We revised figure 2, added the corresponding parameters of the feature layer, adjusted the corresponding relationship between the legend and the legend color, and marked the position of the composite structure in the network. The composite structure of this paper is designed by ourselves. There is no relevant reference. We explain it in the corresponding position in the article.

Author action: We added a description of Figure 2 in the article (Line129-136) The modified Fig.2 is as follows.

original image

Updated image

Reviewer#2, Concern # 5: References and the text are not consistent:[24] is not work of Zhao et al. Lu et al.'s work [25] is not about bridge crack detection as asserted by the authors. Feng et al.'s wok [26] is not about bridge crack detection. However, on can stress that they are correlated problems. Just align the previous work and your statements.

Author response: We gratefully appreciate for your valuable suggestions. Bridge cracks and pavement cracks are both concrete surface cracks. From the perspective of network engineering applications, we expect that the network designed in this paper is not limited to bridge crack detection, so we refer to many similar cracks detection research.

Author action: We have revised the relevant parts of the article to strengthen the preciseness of the expression of the article. (Line 112-121 and Line 579-584)________________________________________

Reviewer#2, Concern # 6: [28] is not about deformable convolutions. Please update all the references in the manuscript and make them consistent to the text. The manuscript is not scrutable with that state.

Author response: We are very sorry for our negligence in this manuscript, we reviewed the address of reference [28] and revised the address in the manuscript. In addition, we reorganized all references and revised 6 references.

Author action: Once again, we reorganized all the reference and revised 6 references in this manuscript. (Line566-567, line579-584 and line588-597)________________________________________

Reviewer#2, Concern # 7: In Section 3.1.1, deformation convolution is illustrated however it does not give any credit to the its original inventors: https://arxiv.org/pdf/1703.06211.pdf Please cite properly. ([27] is not that paper.) 

Author response: We gratefully appreciate for your valuable suggestions. We reviewed the address of reference [27] and revised the address in the manuscript.

Author action: The revised address of reference [27] in the manuscript is highlighted. (Line 588-591)________________________________________

Reviewer#2, Concern # 8: In Section 3.1.2, in line 160, the reference [29] is not about depth separable convolution. The DOI in [29], doi:10.4271/2014-01-0975, is not a paper titled Xception. It is titled as Fatigue Behavior of Stainless-Steel Sheet Specimens at Extremely High Temperatures.

Author response: We gratefully appreciate for your valuable suggestions. We reviewed the address of reference [29] and revised the address in the manuscript.

Author action: The revised address of reference [29] in the manuscript is highlighted. (Line 595-598)________________________________________

Reviewer#2, Concern # 9: WCD dataset does not have any reference, thus not accessible.

NOTE: This manuscript has too many fundamental errors. Please carefully review the paper and then I will be able to read it with a mind in peace.

Author response: Thanks again to the reviewer, we are also aware of this problem. We didn't consider this problem in the process of writing the manuscript. WCD data set is a new data set obtained after merging the existing public data sets (SDNET and CCIC), which is not described too much in the first draft. After the reviewer's reminder, we added this part to the manuscript. At the same time, we checked the manuscript more carefully to ensure that there were no basic errors.

Author action: We added this part to the manuscript and carefully checked the full text of the manuscript. (Line312-317)________________________________________

---

## [Decision Letter · Decision Letter 1]

5 Jul 2022

PONE-D-22-07252R1Bridge Crack Detection Based on Improved Single Shot Multi-Box DetectorPLOS ONE

Dear Dr. He,

Thank you for submitting your manuscript to PLOS ONE. After careful consideration, we feel that it has merit but does not fully meet PLOS ONE’s publication criteria as it currently stands. Therefore, we invite you to submit a revised version of the manuscript that addresses the points raised during the review process.

We look forward to receiving your revised manuscript.

Kind regards,

Ardashir Mohammadzadeh, Phd

Academic Editor

PLOS ONE

Reviewers' comments:

Reviewer's Responses to Questions

**Comments to the Author**

1. If the authors have adequately addressed your comments raised in a previous round of review and you feel that this manuscript is now acceptable for publication, you may indicate that here to bypass the “Comments to the Author” section, enter your conflict of interest statement in the “Confidential to Editor” section, and submit your "Accept" recommendation.

Reviewer #1: All comments have been addressed

Reviewer #2: (No Response)

2. Is the manuscript technically sound, and do the data support the conclusions?

Reviewer #1: Yes

Reviewer #2: Yes

3. Has the statistical analysis been performed appropriately and rigorously? 

Reviewer #1: Yes

Reviewer #2: No

4. Have the authors made all data underlying the findings in their manuscript fully available?

Reviewer #1: Yes

Reviewer #2: Yes

5. Is the manuscript presented in an intelligible fashion and written in standard English?

Reviewer #1: Yes

Reviewer #2: Yes

6. Review Comments to the Author

Reviewer #1: I am satisfied with new changes. authors addressed my all questions except the one which should be updated in the last version.

1. Discuss the stability of the system in terms of complexity.

Reviewer #2: The work has originality; however, it will be in a better state when the following comments are considered.

1. In the lines 234-243, the authors discuss an improved version of the original inception module and illustrates the improved version at Fig5b. Inception-v3 is presented in the “Rethinking the Inception Architecture for Computer Vision” of the Szegedy et. al.’s work. The authors should elaborate their improvement of Fig5b over the Fig5 of Szegedy’s Inception-v3 work. If that network is implied, please reference it.

In the Fig.5b, the network concatenates onto “previous layer”, it should be checked.

2. In section 3.1.3, the marriage of depth wise separable convolutions and deformations convolutions proposed. However, its inner structure and mechanism is not explained and illustrated in a figure other then a box in the Fig2. The mechanism of the proposed combination network should be illustrated and explained other than or additional to lines 212-219.

3. The line 285 reads L(x,c,l,g) while eq. (14) doesn’t have such a term.

4. In Section 4, authors compare performance of their proposed network with a non-SSD based networks such as CrackDFA, however, with a quick search, there is another proposed work of Xu et. al. titled “Automatic Bridge Crack Detection Using a Convolutional Neural Network” which is also non-SSD based method however based on the depth wise separable convolutions. Authors are urged to add more literature on the problem of “bridge crack detection” with other NNs not just SSD and add comparisons also on such papers.

7. PLOS authors have the option to publish the peer review history of their article (what does this mean?). If published, this will include your full peer review and any attached files.

Reviewer #1: No

Reviewer #2: No

---

## [Author Response · Author response to Decision Letter 1]

7 Jul 2022

Reviewer#1, Concern # 1: Discuss the stability of the system in terms of complexity.‎

Author response: We gratefully appreciate for your valuable suggestions. In the experimental part of the manuscript, we added experiments on network computing efficiency and computing power cost, and comprehensively analyzed the relationship between network complexity, computing power consumption and network performance.

Author action: We revised this part of the article and highlighted it. (Line448-466, Table 5 and Table 6).________________________________________

Reviewer#2, Concern # 1: In the lines 234-243, the authors discuss an improved version of the original inception module and illustrates the improved version at Fig5b. Inception-v3 is presented in the “Rethinking the Inception Architecture for Computer Vision” of the Szegedy et. al.’s work. The authors should elaborate their improvement of Fig5b over the Fig5 of Szegedy’s Inception-v3 work. If that network is implied, please reference it. In the Fig.5b, the network concatenates onto “previous layer”, it should be checked.

Author response: Special thanks to you for your good comments. The expression of the relevant contents of the manuscript is not clear and does not highlight the key points. We have modified the relevant contents of the manuscript to make the contents of the manuscript more rigorous.

Author action: We updated the manuscript by modifying the description of this part. (Line 222-238)________________________________________

Reviewer#2, Concern # 2: In section 3.1.3, the marriage of depth wise separable convolutions and deformations convolutions proposed. However, its inner structure and mechanism is not explained and illustrated in a figure other then a box in the Fig2. The mechanism of the proposed combination network should be illustrated and explained other than or additional to lines 212-219.

Author response: Thank you again for your positive comments and valuable suggestions to improve the quality of our manuscript. In view of these problems, we have added the structure diagram of relevant parts. 

Author action: We updated the manuscript by adding the structure diagram of this part. 

FIG 5. Schematic diagram of the depth separable deformable convolution.

Reviewer#2, Concern # 3: The line 285 reads L(x,c,l,g) while eq. (14) doesn’t have such a term.

Author response: Thanks for your careful checks. We are sorry for our carelessness. Based on your comments, we have made the corrections to make the unit harmonized within the whole manuscript.

Author action: We have checked the manuscript again to ensure its quality.

Reviewer#2, Concern # 4: In Section 4, authors compare performance of their proposed network with a non-SSD based networks such as CrackDFA, however, with a quick search, there is another proposed work of Xu et. al. titled “Automatic Bridge Crack Detection Using a Convolutional Neural Network” which is also non-SSD based method however based on the depth wise separable convolutions. Authors are urged to add more literature on the problem of “bridge crack detection” with other NNs not just SSD and add comparisons also on such papers.

Author response: Special thanks to you for your good comments. We added more comparison networks in the comparison experiment, and carried out the comparison experiment under the same experimental platform and experimental settings, which made the performance comparison objects of our network more diverse, not limited to SSD based networks, and increased the persuasion of the manuscript.

Author action: We updated the manuscript by modifying the description of this part. (Line 303-306, 448-466 and 471-512, Table 5, Table 6, Fig. 10 and Fig. 11)________________________________________

---

## [Decision Letter · Decision Letter 2]

2 Sep 2022

PONE-D-22-07252R2Bridge Crack Detection Based on Improved Single Shot Multi-Box DetectorPLOS ONE

Dear Dr. He,

Thank you for submitting your manuscript to PLOS ONE. After careful consideration, we feel that it has merit but does not fully meet PLOS ONE’s publication criteria as it currently stands. Therefore, we invite you to submit a revised version of the manuscript that addresses the points raised during the review process.

We look forward to receiving your revised manuscript.

Kind regards,

Ardashir Mohammadzadeh, Phd

Academic Editor

PLOS ONE

Journal Requirements:

Additional Editor Comments:

Add a direction for readers; add some statements that how the performance can be improved by type-3 fuzzy logic systems;

Reviewers' comments:

Reviewer's Responses to Questions

**Comments to the Author**

1. If the authors have adequately addressed your comments raised in a previous round of review and you feel that this manuscript is now acceptable for publication, you may indicate that here to bypass the “Comments to the Author” section, enter your conflict of interest statement in the “Confidential to Editor” section, and submit your "Accept" recommendation.

Reviewer #2: (No Response)

2. Is the manuscript technically sound, and do the data support the conclusions?

Reviewer #2: Yes

3. Has the statistical analysis been performed appropriately and rigorously? 

Reviewer #2: Yes

4. Have the authors made all data underlying the findings in their manuscript fully available?

Reviewer #2: Yes

5. Is the manuscript presented in an intelligible fashion and written in standard English?

Reviewer #2: Yes

6. Review Comments to the Author

Reviewer #2: Thank you for addressing most of my comments. I have concern about one comment I did in the second revision request (Comment 1). I believe I couldn't be clear about it. I originally mentioned about Szegedy’s Inception-v3 work. There are three models of interest that are portrayed in three figures: Figure X, Figure Y, and Figure Z.

Figure X: Figure 2(b) of Going deeper with convolutions (https://arxiv.org/pdf/1409.4842.pdf).

Figure Y: Figure 5 of Rethinking the Inception Architecture for Computer Vision (https://arxiv.org/pdf/1512.00567.pdf).

Figure Z: Figure 6(b) of this manuscript (previously Figure 5(b)).

Figure X is the original Inception module of Szegedy et. al.'s work, that is authors of this manuscript also indicate an improved version over the Figure X.

Figure Y is an improved version of Figure X which is also proposed by Szegedy et. al.

Figure Z is the improved version of Figure X that authors proposes.

In lines 222-238, the authors beautifully explained how this proposed network works and why those layers are chosen. However, I urge the authors to comment on the improvement of the model in Figure Z over the model in Figure Y. In other words, models in Figure Y and Z should be compared.

7. PLOS authors have the option to publish the peer review history of their article (what does this mean?). If published, this will include your full peer review and any attached files.

Reviewer #2: **Yes: **Ahmet Agirman

---

## [Author Response · Author response to Decision Letter 2]

13 Sep 2022

Original Manuscript ID: PONE-D-22-07252R1 

Original Article Title: Bridge Crack Detection Based on Improved Single Shot Multi-Box Detector

To: PLOS ONE Editor

Re: Response to reviewers

Dear Editor,

Thank you for allowing a resubmission of our manuscript, with an opportunity to address the reviewers’ comments.

We are uploading (a) our point-by-point response to the comments (below) (response to reviewers), (b) an updated manuscript with yellow highlighting indicating changes (Supplementary Material for Review), and (c) a clean updated manuscript without highlights (Main Manuscript).

Best regards,

<Xiaohui He> et al.

Reviewer#1, Concern # 1: Discuss the stability of the system in terms of complexity.‎

Author response: We gratefully appreciate for your valuable suggestions. In the experimental part of the manuscript, we added experiments on network computing efficiency and computing power cost, and comprehensively analyzed the relationship between network complexity, computing power consumption and network performance.

Author action: We revised this part of the article and highlighted it. (Line448-466, Table 5 and Table 6).________________________________________

Reviewer#2, Concern # 1: In the lines 234-243, the authors discuss an improved version of the original inception module and illustrates the improved version at Fig5b. Inception-v3 is presented in the “Rethinking the Inception Architecture for Computer Vision” of the Szegedy et. al.’s work. The authors should elaborate their improvement of Fig5b over the Fig5 of Szegedy’s Inception-v3 work. If that network is implied, please reference it. In the Fig.5b, the network concatenates onto “previous layer”, it should be checked.

Author response: Special thanks to you for your good comments. The expression of the relevant contents of the manuscript is not clear and does not highlight the key points. We have modified the relevant contents of the manuscript to make the contents of the manuscript more rigorous.

Author action: We updated the manuscript by modifying the description of this part. (Line 222-238)________________________________________

Reviewer#2, Concern # 2: In section 3.1.3, the marriage of depth wise separable convolutions and deformations convolutions proposed. However, its inner structure and mechanism is not explained and illustrated in a figure other then a box in the Fig2. The mechanism of the proposed combination network should be illustrated and explained other than or additional to lines 212-219.

Author response: Thank you again for your positive comments and valuable suggestions to improve the quality of our manuscript. In view of these problems, we have added the structure diagram of relevant parts. 

Author action: We updated the manuscript by adding the structure diagram of this part. 

FIG 5. Schematic diagram of the depth separable deformable convolution.

Reviewer#2, Concern # 3: The line 285 reads L(x,c,l,g) while eq. (14) doesn’t have such a term.

Author response: Thanks for your careful checks. We are sorry for our carelessness. Based on your comments, we have made the corrections to make the unit harmonized within the whole manuscript.

Author action: We have checked the manuscript again to ensure its quality.

Reviewer#2, Concern # 4: In Section 4, authors compare performance of their proposed network with a non-SSD based networks such as CrackDFA, however, with a quick search, there is another proposed work of Xu et. al. titled “Automatic Bridge Crack Detection Using a Convolutional Neural Network” which is also non-SSD based method however based on the depth wise separable convolutions. Authors are urged to add more literature on the problem of “bridge crack detection” with other NNs not just SSD and add comparisons also on such papers.

Author response: Special thanks to you for your good comments. We added more comparison networks in the comparison experiment, and carried out the comparison experiment under the same experimental platform and experimental settings, which made the performance comparison objects of our network more diverse, not limited to SSD based networks, and increased the persuasion of the manuscript.

Author action: We updated the manuscript by modifying the description of this part. (Line 303-306, 448-466 and 471-512, Table 5, Table 6, Fig. 10 and Fig. 11)________________________________________

---

## [Editor Report · Decision Letter 3]

19 Sep 2022

Bridge Crack Detection Based on Improved Single Shot Multi-Box Detector

PONE-D-22-07252R3

Dear Dr. He,

We’re pleased to inform you that your manuscript has been judged scientifically suitable for publication and will be formally accepted for publication once it meets all outstanding technical requirements.

Kind regards,

Ardashir Mohammadzadeh, Phd

Academic Editor

PLOS ONE
---

## [Editor Report · Acceptance letter]

23 Sep 2022

PONE-D-22-07252R3 

Bridge Crack Detection Based on Improved Single Shot Multi-Box Detector 

Dear Dr. He:

I'm pleased to inform you that your manuscript has been deemed suitable for publication in PLOS ONE. Congratulations! Your manuscript is now with our production department. 

Kind regards, 

on behalf of

Dr. Ardashir Mohammadzadeh 

Academic Editor

PLOS ONE